# Discrimination of the Lame Limb in Horses Using a Machine Learning Method (Support Vector Machine) Based on Asymmetry Indices Measured by the EQUISYM System

**DOI:** 10.3390/s25041095

**Published:** 2025-02-12

**Authors:** Emma Poizat, Mahaut Gérard, Claire Macaire, Emeline De Azevedo, Jean-Marie Denoix, Virginie Coudry, Sandrine Jacquet, Lélia Bertoni, Amélie Tallaj, Fabrice Audigié, Chloé Hatrisse, Camille Hébert, Pauline Martin, Frédéric Marin, Sandrine Hanne-Poujade, Henry Chateau

**Affiliations:** 1Centre d’Imagerie et de Recherche sur les Affections Locomotrices Equines (CIRALE), Ecole Nationale vétérinaire d’Alfort, 94700 Maisons-Alfort, France; emmapoizat.vet@gmail.com (E.P.); mahaut.gerard@vet-alfort.fr (M.G.); virginie.coudry@vet-alfort.fr (V.C.); amelie.tallaj@vet-alfort.fr (A.T.);; 2LIM France, Labcom LIM-EnvA, 24300 Nontron, France; 3Laboratoire de BioMécanique et BioIngénierie (UMR CNRS 7338), Centre of Excellence for Human and Animal Movement Biomechanics (CoEMoB), Université de Technologie de Compiègne (UTC), Alliance Sorbonne Université, 60200 Compiègne, France

**Keywords:** horse, lameness, symmetry, straight-line, inertial measurement unit, machine learning, support vector machine

## Abstract

Lameness detection in horses is a critical challenge in equine veterinary practice, particularly when symptoms are mild. This study aimed to develop a predictive system using a support vector machine (SVM) to identify the affected limb in horses trotting in a straight line. The system analyzed data from inertial measurement units (IMUs) placed on the horse’s head, withers, and pelvis, using variables such as vertical displacement and retraction angles. A total of 287 horses were included, with 256 showing single-limb lameness and 31 classified as sound. The model achieved an overall accuracy of 86%, with the highest success rates in identifying right and left forelimb lameness. However, there were challenges in identifying sound horses, with a 54.8% accuracy rate, and misclassification between forelimb and hindlimb lameness occurred in some cases. The study highlighted the importance of specific variables, such as vertical head and withers displacement, for accurate classification. Future research should focus on refining the model, exploring deep learning methods, and reducing the number of sensors required, with the goal of integrating these systems into equestrian equipment for early detection of locomotor issues.

## 1. Introduction

Lameness evaluations are a crucial aspect of equine veterinary practice, with their prevalence increasing steadily each year [1]. Typically, these evaluations are conducted visually, with veterinarians observing the horse’s movement. However, this method is inherently subjective [2] and particularly difficult when assessing horses with mild lameness [3]. Moreover, there is a notable disparity in the ability to detect an asymmetrical movement between experienced and inexperienced practitioners [4,5]. Given that detecting lameness can already be challenging for a veterinarian, it becomes even more difficult for an amateur rider. However, early detection of any locomotor abnormality is crucial for animal welfare and preventive care. Identifying such issues as early as possible not only helps prevent unnecessary suffering during work but also allows for more effective and timely treatment. In this context, any assistance in improving detection with tools or expert systems would be highly beneficial. Various tools using IMUs have been developed to quantify lameness, with the goal of objectifying the subjective nature of veterinarians’ visual assessments. IMUs have been proven to be both a reproducible and reliable solution [2,6,7]. However, despite these advantages, the analysis and interpretation of the extensive data and curves generated by these systems remain challenging.

Interpreting movement asymmetry measurements presents several complexities, including determining if a horse is lame and identifying the affected limb. The primary difficulty is synthesizing the vast amount of data into a coherent locomotion assessment. One potential solution is developing an automated method to classify gait and identify the lame limb. Audigié et al. [8] employed a Fourier analysis-based algorithm to identify the lame limb in trotting horses by first classifying the lameness as front or hind and then determining the affected side. However, this study used kinematic methods on a limited number of horses, as the use of IMUs was not yet available at the time. A larger sample size of horses is certainly needed to allow for a more in-depth analysis.

When classifying sound horses, it has been demonstrated that some may exhibit physiological asymmetry while still being considered sound [9], which can result in overinterpretation of lameness when using IMUs. To address this issue, some studies have attempted to establish asymmetry thresholds for more accurate lameness detection.

The most commonly used asymmetry measurement with IMUs is vertical displacement of the head, the pelvis, and most of the time, the withers. In their work, McCracken et al. [3,10,11,12] established asymmetry thresholds for vertical displacement of the head (<6 mm) to detect forelimb lameness and of the pelvis (<3 mm) for hindlimb lameness. However, there are limitations to using absolute values, as they do not account for size variability between horses. Additionally, the use of head vertical displacement may lack accuracy due to compensatory movements in cases of hindlimb lameness [13] and additional head movements [14], particularly in instances of primary lameness, where both the head and pelvis exhibit asymmetrical vertical movements. In such situations, the withers have proven to be a reliable indicator [10,11]. The relationship between head and withers asymmetry can help predict patterns of head and pelvic asymmetry. Horses with contralateral head-withers asymmetry typically show ipsilateral head-pelvic asymmetry, while those with ipsilateral head-withers asymmetry often exhibit contralateral head-pelvic asymmetry [15].

Recently, Macaire et al. [16] explored asymmetry thresholds in horses trotting in a straight line on a hard surface using the EQUISYM® system (Arioneo, LIM France, Nouvelle-Aquitaine, France) with IMU sensors. Based on relative vertical displacement, they found that upward withers amplitudes had the highest accuracy in distinguishing between left and right forelimb lameness, with a sensitivity (Se) greater than 84% and specificity (Sp) over 88%. Similarly, pelvic upward amplitudes and maximum pelvic altitudes were the most effective at identifying left and right hindlimb lameness, with a Se above 78% and Sp over 82%. A common limitation of all these studies on asymmetry measurements is their reliance on a single measurement to differentiate between lame and sound horses.

Schobesberger and Peham [17] were among the first to explore the potential of artificial neural networks (ANN) for equine lameness detection, integrating infrared motion tracking on a treadmill with a multi-layer feedforward ANN trained to classify lameness severity, achieving a correct classification rate of 78.6%. Over the past few years, technological advancements have extended the use of machine learning methods for automatic clinical classification applications [18,19], particularly for gait classification [20,21]. Several classifier models can be used [22,23,24], and have already been applied in the equine veterinary field, with a significant potential [25,26].

For equine gait analysis, Pfau et al. [27] applied linear discriminant analysis (LDA) to automatically classify mild hindlimb lameness using data exclusively from the tuber coxae (dorsoventral and craniocaudal displacement). In their study, LDA, utilizing craniocaudal, mediolateral, and dorsoventral accelerations, velocities, and displacements as input features, achieved a sensitivity (Se) of 100% and a specificity (Sp) of 66% for distinguishing lame from non-lame horses in a sample of 21 subjects. While the sensitivity was high, the specificity could be considered questionable. Additional features remain to be explored, which may improve specificity. Furthermore, the sample size was relatively small for discriminating between left hindlimb, right hindlimb, and sound horses.

The LDA statistical method has certain limitations, including its reliance on strong assumptions about variable distribution and its sensitivity to outliers [22]. Support vector machines (SVM) are widely used for gait analysis and classification [20,28,29], particularly in equine orthopedics for various tasks such as gait classification [30], speed estimation [31], or terrain type detection [32] and have been shown to outperform LDA in several studies [30,33,34]. SVM offers several advantages, including the ability to handle non-linear data structures, reduced sensitivity to overfitting, and often providing greater accuracy compared to other machine learning methods, such as random forests [22,35].

Another statistical method using continuous wavelet transformation has been studied to detect the lame limb, focusing solely on the forelimbs [36].

To the best of our knowledge, no predictive system currently exists that directly identifies the lame limb using multi-variate analysis based on a comprehensive assessment of indices provided by IMUs, without the need to separately analyze forelimb and hindlimb lameness. The aim of this study was to distinguish between right forelimb lameness, left forelimb lameness, right hindlimb lameness, left hindlimb lameness, and sound horses by analyzing vertical displacements (head, withers, pelvis) and other previously unexplored variables such as stride frequency and protraction/retraction angle of the limbs using SVM in horses trotting in a straight line.

## 2. Materials and Methods

This retrospective clinical observational study was approved by the Clinical Research Ethics Committee (ComERC n°2022-01-19).

### 2.1. Horses

This study was conducted on 287 horses brought to the Equine Clinic (CIRALE) at the National Veterinary College of Alfort (Maisons-Alfort, France) for locomotor assessment between April 2019 and February 2023.

### 2.2. Facilities and Locomotor Examination

Following the collection of the anamnesis and the examination of the locomotor system, the veterinarian assessed the horse’s locomotion without a warm-up under the strict conditions of a standard locomotor examination. During the dynamic locomotor evaluation, horses were trotted by their owner or groom along a 25-m straight line on an asphalt surface. The handler was instructed to maintain an appropriate speed and a steady pace. The visual evaluation was performed by one of the five veterinary specialists, each holding the DESV (French certification as a specialist in equine locomotor pathology) and certified by the International Society of Equine Locomotor Pathology (ISELP). Based on these evaluations on a straight line, horses were classified into five groups: right forelimb (RF) lame, left forelimb (LF) lame, right hindlimb (RH) lame, left hindlimb (LH) lame, and sound horses.

In total, 287 horses were included in this study after a locomotor examination. Of these, 256 horses were classified unequivocally lame from a single limb on a straight line at the trot when showing lameness grades of this limb ranging between 2/10 (inclusive) and 7/10 (inclusive) on an 11-grade scale equivalent to the UK scale (where 0 is sound and 10 is non-weight-bearing lameness) [37,38,39]. With these criteria, the final sample comprised 89 LF lame horses, 92 RF lame horses, 36 LH lame horses, and 39 RH lame horses. Horses with multi-limb lameness were not included.

Among the 287 horses, 31 horses were classified as “sound” based on the following criteria: (1) The horses were actively in training, and their owners considered them capable of performing all required exercises at their sport level; (2) three out of five veterinary specialists independently reviewed blinded videos of the horses walking and trotting in a hard circle in both directions, as well as in a straight line on a hard surface, and observed no signs of locomotion abnormalities throughout the video (lameness grade < 1/10).

### 2.3. Data Collection

During the locomotor examination, as part of the clinical routine, horses were systematically equipped with the EQUISYM® system (Arioneo, Nontron, France) as described by Macaire et al. and Timmerman et al. [6,16]. The system is composed of 7 wireless IMUs (tri-axial accelerometer ±16 × gravity, tri-axial gyroscope ±2000 deg/s) placed on the head, the withers, the pelvis, and the 4 cannon bones. The sensors were positioned by a trained operator, and the study by Timmermann et al. [6] demonstrated that there is good inter-operator reproducibility when repositioning the system and repeating the same measurements over successive time intervals. Data were recorded at 200 Hz over approximately two trot-ups, corresponding to an average of 14.7 ± 7.8 trot strides on a straight line. The data were recorded on the sensors and downloaded wirelessly.

### 2.4. Data Processing

The data were processed according to the methods outlined by Macaire et al. and Timmerman et al. [6,16,40]. Briefly, this study utilized asymmetry indices (AsI), based on the vertical displacement of the head (_H), the withers (_W), and the pelvis (_P), during a stride, to compare the left vs. right sides. The AsI values are expressed as a percentage of the maximal range of motion within a stride (Figure 1): AsI-Min, AsI-Max, AsI-Tmax, AsI-up, AsI-down.

In this study, five new features have been added and are described in Table 1.

### 2.5. Data Analysis

To classify the five groups defined in this study (RF, LF, RH, LH, sound), a support vector machine (SVM) with a radial basis function (RBF) kernel was used. All input features were standardized (centered and scaled). To select the optimal model with the best accuracy, a grid search strategy was employed: the cost parameter *c* was tuned across 10 values from 0.25 to 128, doubling with each increment ([0.25, 0.5, 1, …, 64, 128]), while the tuning parameter *sigma* was held constant at 0.036. To avoid bias toward the majority classes, weights were added to the model to balance the predictions across the five classes.

Due to the size of classes, splitting data into train and test samples would introduce bias (with the test sample being too small for the minority classes). Therefore, the model was trained on the entire dataset and validated using bootstrapping [41] with 400 repetitions. This non-parametric approach to statistical inference allows for more accurate estimation of the sample distribution compared to simple replication of the original sample.

The SVM model was implemented using RStudio (RStudio Inc. version 4.2.0, Boston, MA, USA,) and the caret (Classification And REgression Training) library [42].

After selecting the optimal model, the analysis focused on the model’s accuracy using a confusion matrix. An additional analysis was then conducted to evaluate the degree of contribution of each feature (variable) in the generated model.

## 3. Results

### 3.1. Descriptive Results/Description of the Study Population

Data were obtained from a sample of 287 horses, comprising 127 mares, 132 geldings and 28 stallions, with a mean age of 9.3 ± 3.7 years. Among the 256 lame horses, the distribution of lameness across the four limbs and its severity are presented in Table 2.

### 3.2. Accuracy Levels of the SVM Model’s Predictions

The optimal model selected by the grid search strategy was the radial basis function as the kernel type, with a cost value of 1. The class predictions obtained using SVM are presented with a confusion matrix in Table 3.

The SVM prediction model demonstrates an overall accuracy of 86%.

The predicted class was 100% accurate in discriminating between right and left forelimb lameness, as well as right and left hindlimb lameness. The model therefore appears highly effective at lateralizing lameness once it has been identified in front or hind.

The groups with the best predictions are RF, LF, and LH, with accuracy rates of 93.5%, 91%, and 91.7%, respectively. The model predicted the sound horses group with the lowest accuracy, achieving a rate of 54.8%. Additionally, 38.8% of the sound horses were misclassified, with 19.4% categorized as having right forelimb lameness and 19.4% as having left forelimb lameness.

In the group with right hindlimb lameness, a portion was classified as having right forelimb lameness (12.8%) or left forelimb lameness (10.3%), and a small percentage was incorrectly identified as sound.

### 3.3. Degree of Contribution of the Variables in the Model

A total of 27 variables were included in the model, each contributing with varying degrees of importance, as shown in Table 4. The mean values (±standard deviation) obtained for each group are presented in Appendix A.

The most contributive variable for all five groups (sound, RF, LF, RH, LH) was AsI-up. Data related to the pelvis (AsI-up_P) were less discriminative compared to the head (AsI-up_H) and withers (AsI-up_W). Additionally, these two variables contributed less to the sound horses compared to the other four groups. Overall, the importance value was consistently lower for the sound horses group.

AsI-Tmax and AsI-min also contributed effectively to distinguishing between the five groups. However, data from the pelvis for these three variables (AsI-up_P, AsI-min_P, AsI-Tmax_P) contributed less.

Vertical head measurement data made an important contribution to the model, including the AsI-down feature, which, in contrast, did not contribute as significantly when using data from the withers and pelvis. Additionally, some of the newly included variables, such as ROM, ERz, and stride frequency, also showed limited contribution to the model’s performance.

The other indices (Δϕ, AsI-retraction, AsI-max) yielded poor contributions. Stride frequency did not play a significant role in the classification of the five groups. Similarly, the features “range of motion” and "ERz" showed limited contribution to the model’s performance, regardless of whether the data were derived from the head, withers, or pelvis.

## 4. Discussion

### 4.1. Lameness Identification

This study tested the hypothesis that a machine learning method can synthesize data provided by IMU measurements into a locomotion assessment, accurately identifying the lame limb. The SVM model was chosen as a proven method but was not compared to other models. The confusion matrix obtained with the model developed illustrates the classification of the horses in the sample into five categories (sound, RF, LF, RH, and LH lameness). This matrix not only visualizes the model’s errors but also allows for a deeper analysis of how these errors occur and the contribution of the retained variables in the model, providing valuable insights for improvement.

Among the four lame limbs, the right hindlimb group was the least accurately predicted (76.9%), with most misclassifications occurring as right and left forelimb lameness. This may be attributed to the smaller sample size of hindlimb lameness cases (75) compared to forelimb lameness cases (181). Additionally, the RH group had the fewest instances of high-grade lameness (only 3 horses had a grade ≥ 5/10), which may have contributed to the model’s increased confusion.

When front and hindlimb lameness were identified, the model never confused right and left lameness. Conversely, identifying whether lameness originates in the forelimbs or hindlimbs can be challenging due to compensatory movements, which may be influenced by the location and underlying cause of the lameness [43]. In our reference sample and with the developed model, confusion between forelimb and hindlimb lameness occurred in 1.1% to 2.2% of cases where forelimb lameness was misclassified as hindlimb lameness, and in 2.8% to 12.8% of cases where hindlimb lameness was misclassified as forelimb lameness. Once again, increasing the sample size for hindlimb lameness and including more cases of high-grade lameness (≥5/10) could help improve the model’s ability to differentiate between forelimb and hindlimb lameness.

Previous studies have explored this topic using various data analysis methods, with some focusing only on forelimb lameness [36] and others on hindlimb lameness [27]. Audigié et al. [8] examined a small sample (n = 25) that included both forelimb and hindlimb lameness. However, their classification algorithm was applied only to lame horses, and the challenge of distinguishing between physiological asymmetry and true lameness remains. The present study aimed to classify both lame and sound horses. One of the main limitations, however, was the difficulty in accurately identifying non-lame individuals, who were the least accurately predicted by the model, with a rate of 54.8%.

This confirms that establishing thresholds for a large population is challenging, as they may exhibit physiological degrees of asymmetry. Sound horses were largely misclassified as having forelimb lameness (38, 8%). This may be attributed to the fact that horses sometimes display additional head movements [9,14,44,45,46], which can disrupt the model’s performance. Furthermore, horses classified as sound may still exhibit some degree of asymmetry [9,47,48], potentially leading to overinterpretation of lameness and raising the key question: is it physiological asymmetry or pathological lameness? This highlights the need for a clinical consensus, which is difficult to establish and can only be achieved through human expertise, based on experience. This is an important consideration, a point to consider when analyzing misclassified individuals, in contrast to those correctly classified with good accuracy (86%). This observation has been noted in other studies on lameness classification using IMUs [49]. Pfau et al. [27] achieved 100% sensitivity and 66% specificity in distinguishing hindlimb lameness from sound horses in a sample of 21 horses. The good sensitivity achieved confirms previous studies on objective assessments, which have generally shown better sensitivity compared to subjective evaluations, especially in cases of mild lameness [3,5,50].

Another important consideration is how the group of sound horses was selected and defined in our database. These horses were assessed by specialist veterinarians using blinded video footage rather than being evaluated in real-life conditions. The small number of horses in the sound group (n = 31) is also a limitation worth noting. Additionally, as mentioned earlier, hindlimb lameness cases represent a small portion of the sample. This results in an imbalance between the five groups. While class weights were adjusted to address this, applying the presented model to a larger and more balanced population would be beneficial.

Earlier studies focused their analyses on the symmetry of head movements [14,51], specific parts of the trunk [52,53,54,55], or the tuber coxae [27]. For this reason, Audigié et al. [8] combined the analysis of cranial and caudal trunk movement symmetries. However, the identification of the lame limb in their approach required a two-step process, as it was not fully automated. In contrast, the present study included 27 variables in the SVM model to ensure more comprehensive and relevant data for a more automated and robust classification.

### 4.2. Analysis of the Contribution of the Variables to the Model

As expected, features related to head vertical displacements are valuable for discriminating forelimb lameness [16,36,56,57]. Indeed, the head indices (AsI-up_H, AsI-min_H, AsI-Tmax_H) contributed the most to classifying the sample into five groups, both for identifying the lame limb and distinguishing sound horses. This indicates that, despite potential random movements [9,14,44,45,46], head indices provide critical information for determining whether a horse is lame and identifying the affected limb.

Withers features (AsI-up_W, AsI-min_W, AsI-Tmax_W, Δϕ_H) made a significant contribution in the model to distinguishing sound horses from lame ones. This is consistent with previously published studies highlighting the importance of considering withers movements, rather than focusing solely on head movements, for determining the location of lameness [10,11,15,58,59]. In contrast, while previous studies have identified the pelvis’s upward movement (AsI-up_P) as the most effective indicator for discriminating hindlimb lameness [16,60], its contribution in this model was surprisingly low. This raises the question of whether the model could function effectively without pelvis data, which could be advantageous in reducing the number of sensors and simplifying the device—particularly in the context of a streamlined system for rider use. However, more in-depth studies are needed to explore this question further. In any case, pelvis data should be considered holistically, rather than solely for hindlimbs. These results demonstrate that the head, withers, and pelvis provide complementary information about both the presence and location of lameness.

In the present model, we included the ROM variable to account for the fact that, depending on the horse (size, gait type, dynamism), the amplitude of vertical movements could vary between individuals and potentially influence classification. In fact, the ROM variable contributed very little to the model, suggesting that this adjustment may not be necessary for improving classification.

The present study confirms the findings of Macaire et al. [16] regarding the AsI-down variable, which here contributed very little to identifying either the lame limb or sound horses. This suggests that AsI-down may not be a particularly valuable feature for these classifications. However, the same cannot be said for AsI-down measured with the head sensor, which had a significantly higher impact—78% for identifying the lame limb and 66.9% for distinguishing sound horses.

The retraction angles were added to the model to account for limb orientation at foot lift-off, as certain locomotor anomalies may manifest as propulsion defects, which can result in reduced retraction. Forelimb retraction data moderately helped the model identify the lame limb but contributed very little to identifying sound horses. This could be due to the large number of mild lameness cases in our sample and the fact that retraction defects are not systematic across all types of lameness. On the other hand, retraction data from the hindlimbs played a more important role in identifying sound horses, as well as RF, LF, and LH lameness. However, the contribution of this feature for identifying RH lameness was surprisingly low compared to the other four classes, making the interpretation of this result more difficult, as it seems quite improbable that such a marked difference between right and left has real clinical significance.

Delta ϕ had a quite strong impact on the model for the head, a moderate effect for the withers, and a minimal effect for the pelvis. Similarly, ERz features did not contribute effectively.

These results highlight the importance of the features included in the model, as each one contributes differently. One limitation regarding the contribution of the variables is that the sample includes all types of lameness without further classification. It is evident that certain indicators are more or less affected depending on the type of injury.

### 4.3. Limitations and Future Works

Our study confirms the hypothesis that a support vector machine method can discriminate, with a global accuracy of 86%, between RF, LF, RH, and LH lameness, as well as sound horses, in horses trotting in a straight line. However, the specificity needs improvement to reduce false positives for horses considered sound, which reignites the ongoing debate about the clinical interpretation of what constitutes a sound horse.

The present results focus solely on single-limb lameness and represent a preliminary step toward developing a predictive system for identifying the lame limb, which would display the probability of accuracy as a percentage. In future work, other machine learning methods could be explored and tested, as well as approaches to reduce the amount of input data, simplifying capture systems by reducing the number of sensors. The ultimate goal would be to integrate these systems into equestrian equipment, enabling the implementation of alerts for the riders.

Methods using raw signals rather than aggregated features as input can be beneficial to keep all information contained in the temporal signals. They can also be more easily transferred to other settings and tasks. Such methods rely on deep learning and often need large datasets and could be of particular interest, especially if more hindlimb lameness cases and sound horses are added to the database. Deep learning models have been applied to predict vertical ground reaction forces [61], classify gaits [30], or classify forelimb lameness [62], but they have not yet been used to predict the lame limb overall.

## 5. Conclusions

This study established a method for predicting the lame limb using a support vector machine in horses trotting in a straight line. Right forelimb, left forelimb, and left hindlimb lameness are predicted with rates above 91%. Right hindlimb lameness has a lower rate of 76.9%, but the most difficult class to predict is the sound one with a 54.8% rate. Our study also highlights the contribution of various variables in the model, helping to better understand their significance. In particular, head and withers variables (AsI-up_W, AsI-min_W, AsI-Tmax) were especially important for the model, and pelvis variables were less used. It is important to note, however, that horses with multi-limb lameness were excluded, and this limitation will need to be addressed in the future to continue advancing this analysis.

Future studies utilizing deep learning methods are needed to continue improving and developing automatic systems that accurately classify gait to identify the lame limb, with the ultimate goal of providing riders with external, automated tools for early detection of locomotor anomalies and improving their management.

## Figures and Tables

**Figure 1 sensors-25-01095-f001:**
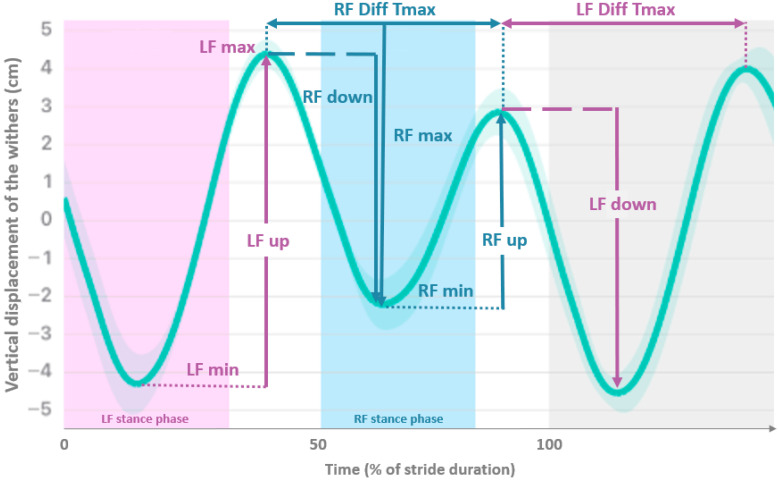
Mean vertical displacement (cm) of the withers plotted against time (expressed as a percentage of stride duration) of a horse showing right forelimb (RF) lameness. Asymmetry Indices (AsI) are AsI-min = RFmin − LFmin/LFup; AsI-max = LFmax − RFmax/LFup; AsI-up = LFup − RFup/LFup; AsI-down = LFdown − RFdown/LFdown and AsI-Tmax = LFdiffTmax − RFdiffTmax/LFdiffTmax. (LF-Left Forelimb).

**Table 1 sensors-25-01095-t001:** Description of the features added in this study.

Features	Abbreviation	Meaning
AsI Retraction angle (%)	AsI-retraction	Percentage of left-right difference of the angles of the metacarpus sensor relative to the vertical at the moment of foot lift off
Energy ratio (%)	ERz	Ratio of energy between the second and first harmonics of the Fourier decomposition of the signal, as described by Audigié et al. [8]
Delta ϕ (°)	Δϕ	Difference in phase harmonic value between first and second harmonics of the signal, as described by Audigié et al. [8]
Stride frequency (Hz)	SF	Number of trotting strides per second
Maximal Range Of Motion (cm)	ROM	Maximum amplitude of vertical movement

**Table 2 sensors-25-01095-t002:** Number of horses showing a lameness depending on the location and the grade according to the 11-grade UK lameness scale. Mean ± SD ^1^ of lameness grade in each group of lame horses.

Lameness Grade	2/10	3/10	4/10	5/10	6/10	7/10	Total	Mean ± SD ^1^
Left Forelimb lameness	53	13	13	5	3	2	89	2.9 ± 1.3
Right Forelimb lameness	51	12	23	4	2	0	92	2.8 ± 1.1
Left Hindlimb lameness	13	8	8	1	6	0	36	3.4 ± 1.4
Right Hindlimb lameness	18	4	14	1	2	0	39	3.1 ± 1.2

SD ^1^—standard deviation.

**Table 3 sensors-25-01095-t003:** Confusion matrix of class predictions. The bold values highlight the number and percentage of horses for which the model’s predicted class matches the actual class.

	Predicted	RF Lame	LF Lame	RH Lame	LH Lame	Sound
Actual						
RF lame		**86 (93.5%)**	0 (0%)	0 (0%)	2 (2.2%)	4 (4.3%)
LF lame		0 (0%)	**81 (91.0%)**	1 (1.1%)	1 (1.1%)	6 (6.7%)
RH lame		5 (12.8%)	4 (10.3%)	**30 (76.9%)**	0 (0%)	0 (0%)
LH lame		1 (2.8%)	2 (5.6%)	0 (0%)	**33 (91.7%)**	0 (0%)
Sound		6 (19.4%)	6 (19.4%)	1 (3.2%)	1 (3.2%)	**17 (54.8%)**

**Table 4 sensors-25-01095-t004:** Variable importance in the SVM model for distinguishing sound horses and lame horses with right forelimb (RF), left forelimb (LF), right hindlimb (RH), and left hindlimb (LH) lameness, expressed as a percentage (from 0% not used at all, to 100%, highly important), for the head (_H), withers (_W), and pelvis (_P). A continuous color gradient is used to represent variable importance: deep green: 100% importance; white: 60% importance; deep red: 0% importance.

	Straight Line				
	Sound	RF	LF	RH	LH
AsI-up_H (%)	89.1	97.2	97.2	97.2	97.2
AsI-up_W (%)	89.5	100.0	100.0	100.0	100.0
AsI-up_P (%)	89.0	89.0	74.1	63.3	63.3
AsI-down_H (%)	66.9	78.0	78.0	78.0	78.0
AsI-down_W (%)	30.5	37.0	37.0	37.0	37.0
AsI-down_P (%)	31.0	31.0	25.9	25.9	25.9
AsI-max_H (%)	59.0	70.8	70.8	70.8	70.8
AsI-max_W (%)	74.0	74.0	72.1	72.1	72.1
AsI-max_P (%)	87.5	87.5	60.8	60.8	60.8
AsI-min_H (%)	89.2	95.2	95.2	95.2	95.2
AsI-min_W (%)	81.2	91.7	91.7	91.7	91.7
AsI-min_P (%)	67.8	66.4	67.8	38.0	39.3
AsI-retraction_W (%)	38.2	63.2	63.2	63.2	63.2
AsI-retraction_P (%)	77.9	77.9	65.2	2.2	70.1
AsI-Tmax_H (%)	87.7	94.3	94.3	94.3	94.3
AsI-Tmax_W (%)	71.9	89.7	89.7	89.7	89.7
AsI-Tmax_P (%)	68.4	66.5	68.4	41.6	41.6
ERz_H (%)	38.4	38.4	18.6	54.4	49.1
ERz_W (%)	47.9	47.9	7.5	62.7	58.3
ERz_P (%)	63.3	50.9	63.3	25.9	41.1
Δϕ _H (°)	73.0	76.5	76.5	76.5	88.2
Δϕ _W (°)	56.0	64.3	64.3	64.3	64.3
Δϕ _P (°)	36.1	1.6	36.1	4.4	1.8
Stride frequency (Hz)	20.0	0.0	20.0	19.5	1.0
ROM_H (cm)	8.7	8.7	1.1	43.6	16.4
ROM_W (cm)	21.1	21.1	3.8	41.3	35.1
ROM_P (cm)	51.6	31.5	51.6	13.6	20.6

## Data Availability

The data that support the findings of this study are available from the corresponding author upon reasonable request.

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
