# Peer review of "Discrimination of the Lame Limb in Horses Using a Machine Learning Method (Support Vector Machine) Based on Asymmetry Indices Measured by the EQUISYM System"

_sensors, 2025, doi:10.3390/s25041095_

Round 1
Reviewer 1 Report
Comments and Suggestions for Authors
1. In this study, only svm model is used to classify the data. In the experimental part, some comparative experiments can be added to prove that the performance of this model is better than other models.
2. The meaning of the wavy line under the text in Figure 1 needs to be explained in the paper. If the wavy line is meaningless, please delete it.
3. The meanings of colors in Table 4 can be described in the notes. It is recommended to describe the value range represented by each color. (What does it mean from deep to shallow)
Reviewer 2 Report
Comments and Suggestions for Authors
The manuscript reports SVM by equisys system, the logic is fine for readers. The following are the things in need to solve.
1, Why SVM, what is the news on this method? How compared the similar work? Please show them in manuscript
2, I do not agree 54.8%, why this is new?
3, The introduction section is too long, please shorten it
4, Please polish figure 1, there is marks below stride duration in figure
5, How obtain the data? Any new on the system, for example, the sensor system? How instalation of the sensors
6, The discussion component is not organize well, please polish it
7, The conclsion section is not organize well, please insert some number to show the advantage of this research.
8, Please remove preliminary from conclision section. The word is not suitable there.
9, Plenty of papers should cite to support the backhround, such as
Deep Learning Enabled Triboelectric Smart Socks for IoT Based Gait Analysis and VR Applications. npj Flexible Electronics. 2020, 4:29.
Comments on the Quality of English LanguageIt is fine for me
Round 2
Reviewer 1 Report
Comments and Suggestions for Authors
1. A more detailed description of the research about svm can be given in the introduction, especially in the research of horses
